# IMPROVING LLM PREDICTIONS VIA INTER-LAYER STRUCTURAL ENCODERS

**Tom Ulanovski**[*]
Blavatnik School of Computer Science
Tel Aviv University
`tomulanovski@mail.tau.ac.il`

**Eyal Blyachman**[*]
Tel Aviv University
`blyachman1@mail.tau.ac.il`

**Maya Bechler-Speicher**
Meta
`mayabs@meta.com`

## ABSTRACT

Standard LLM inference relies on final-layer representations, despite evidence that intermediate layers often capture task-specific information more effectively. However, identifying the optimal layer remains task-dependent and computationally expensive. In this work we introduce Inter-Layer Structural Encoders (ILSE), an approach to learn one representation from the LLM's internal layer representations all together. Central to ILSE is Cayley-Encoder, a geometric encoder which builds upon recent studies leveraging Cayley Graphs for neural information propagation. We evaluate ILSE across 13 classification and semantic similarity tasks with 2 pre-trained LLMs. ILSE consistently outperforms baselines and existing approaches, achieving up to 40% improvement in accuracy and 22% in similarity metrics.

## 1 INTRODUCTION

While standard practice utilizes the final-layer representation of Large Language Models (LLMs) for downstream tasks, Wallat et al. (2020) and Skean et al. (2025) demonstrated that intermediate layer representations contain substantial task-relevant information and can often outperform final-layer representations, while the optimal layer remains task-dependent. A core question is therefore how to effectively combine representations from all layers.

Existing aggregation methods, such as ELMo's scalar weighting (Peters et al., 2018) or depth-wise attention (ElNokrashy et al., 2024), either lack the capacity for complex interactions or over-relate on the final layer instead of treating all layers equally.

In this work, we propose *Inter-Layer Structural Encoders* (ILSE), an efficient and effective method for combining layer representations from all the layers of the LLM. We treat representation aggregation learning as a structured communication problem. Central to our approach is the **Cayley-Encoder**, which maps layer representations to nodes in a Cayley graph over the Special Linear Group $SL(2, \mathbb{Z}_n)$. These graphs are 4-regular and bottleneck-free, ensuring efficient information flow with logarithmic diameter, properties shown to mitigate over-squashing in GNNs (Alon & Yahav, 2021; Wilson et al., 2025).

We evaluate ILSE across 13 classification and semantic similarity (STS) tasks using 2 LLMs. Our results demonstrate that ILSE consistently outperforms existing baselines, achieving up to 40% accuracy gains. Crucially, ILSE is highly efficient, requiring only 32 samples per label to outperform Last-Layer and Best-Layer baselines in few shot learning. Our code is available at: `https://anonymous.4open.science/r/ILSE-9956/README.md`.

---

[*]Equal contribution

## 2 RELATED WORK

**Inter-Layer Representations.** Transformer layers capture linguistic features hierarchically, with surface features in lower layers, syntactic features concentrate in middle layers, and semantic features more spread across layers with a focus in upper layers (Jawahar et al., 2019; Tenney et al., 2019). Recent studies confirm that intermediate layers often outperform the final layer on downstream tasks (Wallat et al., 2020; Skean et al., 2025). Existing aggregation strategies include linear scalar weighting (Peters et al., 2018) or depth-wise attention (DWAtt) (ElNokrashy et al., 2024). However, scalar methods lack non-linear expressivity, while DWAtt over-rely on the final layer as the sole query. Other architectural modifications (Pagliardini et al., 2024; Xiao et al., 2025) require expensive re-training. In contrast, ILSE enables non-linear, structured communication within a *frozen* transformer.

**Graph Neural Networks and Cayley Graphs.** Graph Neural Networks (GNNs) operate by iteratively propagating information between nodes (Gilmer et al., 2017). Standard GNNs suffer from *over-squashing*, where information is exponentially compressed in fixed-sized node vectors as it propagates through the graph, creating bottlenecks (Alon & Yahav, 2021). Deac et al. (2022) proposed Expander Graph Propagation (EGP) to mitigate this using Cayley graphs. Cayley graphs are sparse, highly connected structures with logarithmic diameter, allowing any two nodes to communicate in $O(\log |V|)$ hops. EGP constructs a Cayley graph and alternates message passing between the original input graph and the Cayley graph. Cayley graphs come in fixed sizes determined by the underlying group, so EGP truncates them to match the input graph size. However, Wilson et al. (2025) showed that truncation can reintroduce bottlenecks and proposed padding the input graph with virtual nodes to preserve the complete Cayley structure, thereby avoiding the bottlenecks introduced by truncation.

## 3 INTER-LAYER STRUCTURAL ENCODERS

We introduce ILSE, a framework for learning a unified representation from all $L$ layers of a frozen LLM. For each layer $\ell$, we compute a representation $\mathbf{z}_\ell \in \mathbb{R}^d$ via mean-pooling over all token representations. To aggregate these, we propose three structural encoders:

**Cayley-Encoder:** We randomly map layers representations to nodes in a 4-regular Cayley graph over the Special Linear Group $SL(2, \mathbb{Z}_n)$. These graphs are high-expansion, bottleneck-free structures with logarithmic diameter, ensuring efficient global information flow (Wilson et al., 2025). The node count is given by:

$$|V_n| = n^3 \prod_{\text{prime } p|n} \left( 1 - \frac{1}{p^2} \right) \tag{1}$$

We choose the smallest $n$ such that $|V_n| \geq L$. To maintain the graph's expansion properties, we pad the system with $|V_n| - L$ *virtual nodes* (initialized as zero vectors). Message passing occurs across the full Cayley structure, but only the $L$ non-virtual nodes are used for the final task-specific representation.

**Set-Encoder:** We treat the layers as an unordered set, utilizing a DeepSet architecture (Zaheer et al., 2017) to learn permutation-invariant aggregation. This serves as a information fusion structure without explicit inter-layer interaction.

**Fully-Connected (FC) Encoder:** We construct a complete graph $K_L$ where every layer is connected to every other layer. This allows a GNN to learn direct pairwise interactions across the entire depth of the LLM.

## 4 EXPERIMENTS

In this section we evaluate ILSE on 5 classification and 8 STS tasks from MTEB (Muennighoff et al., 2023).

Our experiments include: (i) Full-data training on all available task data; (ii) Few-shot training with 1–1024 samples per label; and (iii) Zero-shot STS transfer where we evaluate performance across 7 tasks after training on a single STS task.

Table 1: Performance comparison of ILSE and baselines across 2 different LLMs on 5 classification tasks. In all 10 cases ILSE gets the highest score. For Pythia-410 FC-Encoder is the most dominant method while in Llama3 it's Cayley-Encoder. In all cases but one ILSE also gets the best second and third scores. **Bold**: best per column, blue: 2nd best, red: 3rd best.

| Base Model | Section | Method | Banking77 | Emotion | MTOPDomain | MTOPIntent | PoemSentiment |
|---|---|---|---|---|---|---|---|
| Pythia 410m | Baselines | Last Layer | 61.17 | 33.48 | 80.88 | 66.97 | 42.40 |
| | | Best Single Layer | 66.67 | 35.02 | 83.78 | 71.18 | 45.67 |
| | | MLP Last Layer | 83.84 | 33.99 | 96.97 | 83.75 | 75.00 |
| | | MLP Best Layer | 41.93 | 25.41 | 87.25 | 70.79 | 53.94 |
| | | Weighted | 58.50 | 28.26 | 79.79 | 61.68 | 42.60 |
| | | DWATT | 83.23 | 58.60 | 98.03 | 91.66 | 70.87 |
| | ILSE | Set-Encoder | 84.23 | 47.89 | 97.59 | 92.21 | 73.37 |
| | | FC-Encoder (GIN) | 90.10 | 73.36 | 98.68 | 94.32 | 69.90 |
| | | FC-Encoder (GCN) | **90.65** | **75.61** | 98.65 | **95.04** | **75.77** |
| | | Cayley-Encoder (GIN) | 89.12 | 73.83 | **98.77** | 94.72 | 70.87 |
| | | Cayley-Encoder (GCN) | 89.43 | 66.40 | 98.67 | 94.19 | 69.13 |
| Llama3 8B | Baselines | Last Layer | 68.25 | 34.23 | 84.42 | 73.39 | 40.96 |
| | | Best Single Layer | 71.93 | 38.42 | 89.01 | 78.17 | 47.02 |
| | | MLP Last Layer | 86.70 | 67.67 | 98.58 | 92.09 | 75.00 |
| | | MLP Best Layer | 49.59 | 21.17 | 47.64 | 60.04 | 35.67 |
| | | Weighted | 66.63 | 27.94 | 83.85 | 71.78 | 37.79 |
| | | DWATT | 90.04 | 66.55 | 97.97 | 92.41 | 75.00 |
| | ILSE | Set-Encoder | 87.62 | 71.04 | 98.77 | 95.43 | 77.02 |
| | | FC-Encoder (GIN) | 92.10 | 71.64 | 98.77 | 95.65 | 75.96 |
| | | FC-Encoder (GCN) | 92.38 | 71.03 | 98.99 | 96.19 | 77.98 |
| | | Cayley-Encoder (GIN) | 92.46 | 71.58 | 98.98 | **96.46** | 76.54 |
| | | Cayley-Encoder (GCN) | **92.85** | **73.43** | **99.03** | 95.90 | **79.04** |

**Baselines.** To evaluate ILSE, we compare against: **(1) Last-Layer:** last-layer representation, as done in (Skean et al., 2025); **(2) Best-Layer:** the best-performing layer for each task, selected by evaluating each layer independently; **(3) Weighted:** a learned weighted sum of the layer representations, similarly to (Peters et al., 2018); **(4) MLP Best-Layer:** an MLP trained over the last-layer representations; **(4) MLP Last-Layer:** an MLP trained on the selected layer from Best-Layer; and **(5) DWAtt:** a 256-dim projection followed by depth-wise attention (ElNokrashy et al., 2024). The projections is meant to manage parameter overhead (See Appendix 3).

We evaluate ILSE with two LLM families: Pythia-410M (25 layers, 1024-dim) (Biderman et al., 2023) and Llama3-8B (33 layers, 4096-dim) (AI@Meta, 2024). The base models remain frozen during training and we only train ILSE and the baselines on top of the frozen representations. For each layer, we use the layer representation obtained by mean-pooling over all token representations. These pooled representations serve as the input to all methods and baselines evaluated in this work.

**Tasks.** We evaluate on two types of tasks from MTEB benchmark (Muennighoff et al., 2023): classification and Semantic Text Similarity (STS) (see STS details in the Appendix A).

For the classification tasks we used Banking77Classification (77 banking intent classes of online banking queries), EmotionClassification (six emotions classes of english Twitter messages), MTOP-DomainClassification (domain prediction for task-oriented dialogue inputs), MTOPIntentClassification (user intent prediction for task-oriented dialogue inputs), and PoemSentimentClassification (4-class sentiment analysis of poem verses) (Casanueva et al., 2020; Saravia et al., 2018; Li et al., 2021; Sheng & Uthus, 2020; Enevoldsen et al., 2025). We used the training split for each task to train the model and then evaluated on the test split.

**Setup** For classification tasks, we train the encoders jointly with a linear classifier head using cross-entropy loss. The base LLM remains frozen throughout training. We use the Adam optimizer (Kingma, 2014) with learning rate and weight decay selected via Optuna (Akiba et al., 2019) hyperparameter optimization on the validation set. For FC-Encoder and Cayley-Encoder, we evaluate two GNN architectures: GIN and GCN. We test different parameters values. For the number of MPNN layers we test 1-2, for dropout we test values 0.0 - 0.3, for learning rate and weight decay we test $10^{-4}$ to $10^{-3}$. For the number of layers in the MLP used inside the GIN aggregation we test 1-2. We keep hidden dimension fixed to 256, and batch size to 64 and 256 in classification and STS tasks respectively. We choose the best parameters set based on the best validation accuracy and train the final model. For the test evaluation we extract the trained encoder architecture without the classifier

head for classification and use it for encoding as part of MTEB evaluation - we report the final accuracy and similarity scores on the test set.

We further examine the performance of ILSE in a few-shot learning setting on the classification tasks using Pythia-410M model.

## 4.1 RESULTS

Table 1 presents our main results on classification tasks (see Appendix A for STS results). ILSE achieves the best performance in all 10 classification configurations.

ILSE achieves average gains of +28% over Last-Layer baseline and +24% over the Best-Layer baseline, which itself requires evaluating every layer independently. We observe particularly large margins on EmotionClassification, where Cayley-Encoder improves +39% and +37% over Last-Layer and Best-Layer respectively on average across the two LLMs. Similar patterns emerge across Banking77 and the MTOP tasks, where ILSE consistently reaches 90–99% accuracy compared to 60–85% for Last-Layer and Best-Layer baselines.

Comparing ILSE to multi-layer approaches reveals clear advantages. ILSE outperforms DWAtt on all tasks, with improvements reaching +10% on EmotionClassification, while using roughly 3-5× fewer parameters than DWAtt (Table 3).

Among ILSE encoders, we observe that explicit graph structures (FC-Encoder and Cayley-Encoder) consistently outperform Set-Encoder, indicating that inter-layer connectivity improves aggregation. In classification tasks FC-Encoder and Cayley-Encoder show comparable performance (Cayley-Encoder is particularly effective for STS tasks - see Appendix A).

## 4.2 FEW-SHOT LEARNING

We evaluated few-shot performance by restricting the training data to a range of 1 to 1024 samples per label across all classification tasks (for some tasks where training data is limited, we reach the full training data after 128 samples per label). Our results indicate data efficiency: for 4 tasks (Banking, MTOP Domain, MTOP Intent and Poem Sentiment) ILSE achieves gains over both the Last-Layer and Best-Layer baselines with as few as 8 samples per label. In Banking task using just 32 samples per label makes Cayley-Encoder and FC-Encoder to surpass all other baselines including DWAtt being trained with full amount of training data. The overall trend suggests that ILSE is effective in low-resource settings. This makes it particularly valuable for practical applications where labeled data is scarce (Figure 1).

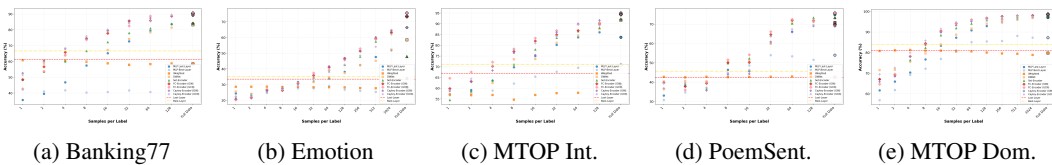

| (a) Banking77 | (b) Emotion | (c) MTOP Int. | (d) PoemSent. | (e) MTOP Dom. |

Figure 1: Few-Shot Learning Analysis. Performance across 1-1024 samples per label using Pythia-410M. ILSE outperforms baselines with 32 samples per label across all tasks.

## 5 CONCLUSION

In this work, we introduced ILSE - three structured layer-fusion strategies: Empty Graph (Deep Sets), Fully-Connected Graph, and Cayley Graph to enhance intermediate representation aggregation in LLMs. Our results demonstrate that structured, graph-based fusion outperforms several baselines. Specifically, ILSE achieves improvements of up to +28% across models and tasks over the Last-Layer representation and +24% over the Best-Layer representation selected after evaluation across all layers. These gains are achieved with ∼400K - 1.2M trainable parameters, representing a 0.015–0.1% of the base model's size. We conclude that Cayley-Encoder achieves the highest performance across our experiments. It offers a theoretically-grounded and efficient approach that leverages all model layers to produce enhanced representations for improved downstream task performance.

## 6 ACKNOWLEDGEMENT

This study was supported in part by a fellowship from the Edmond J. Safra Center for Bioinformatics at Tel-Aviv University.

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

## A    STS Training and Results

For STS tasks, we used STSBenchmark (May, 2021) training split and evaluate on its test split. We then use the same trained models for zero-shot evaluation on all other English STS tasks in MTEB: STS12, STS13, STS14, STS15, STS16, BIOSSES, and SICK-R (Soğancıoğlu et al., 2017; Agirre et al., 2012; 2013; Bandhakavi et al., 2014; Biçici, 2015; Nakov et al., 2016; Marelli et al., 2014).

We encode each sentence in a pair through the frozen LLM and our encoders, compute the cosine similarity between the resulting representations, and minimize Mean Squared Error (MSE) loss with respect to the ground-truth similarity score.

**Results**. ILSE achieves the best performance in the 2 supervised STS configurations, and 11 out of 14 zero-shot STS transfer settings 2.

For STS tasks, Cayley-Encoder improves upon Last-Layer and Best-Layer in 6 out of 8 tasks and is the only method to outperform both Last-Layer and Best-Layer baselines on average.

Cayley-Encoder proves particularly effective for STS tasks, achieving the best score in 6 out of 8 tasks. This suggests that the sparse, regular connectivity of Cayley graphs is particularly beneficial for semantic similarity. The choice between GIN and GCN aggregation shows task-dependent variation, with GIN generally favoring STS tasks and GCN showing slight advantages on classification.

Table 2: Performance comparison of ILSE and baselines across 2 different LLMs on 8 STS tasks. **Bold**: best per column, blue: 2nd best, red: 3rd best.

| Base Model | Section | Method | STSBenchmark | STS12 | STS13 | STS14 | STS15 | STS16 | BIOSSES | SICK-R |
|---|---|---|---|---|---|---|---|---|---|---|
| Pythia 410m | Baselines | Last-Layer | 39.12 | 46.96 | 47.00 | 41.45 | 49.32 | 50.37 | 67.30 | 52.55 |
| | | Best-Layer | 53.53 | 50.62 | 59.27 | 51.61 | 65.59 | 58.02 | **74.80** | **58.26** |
| | | MLP Last-Layer | 25.54 | 42.14 | 36.16 | 33.28 | 37.75 | 39.84 | 59.68 | 43.23 |
| | | Weighted | 41.81 | 47.12 | 49.49 | 44.21 | 53.72 | 52.76 | 67.79 | 55.09 |
| | | DWAtt | 49.40 | 52.51 | 44.57 | 47.46 | 52.51 | 44.80 | 45.29 | 52.43 |
| | ILSE | Set-Encoder | 39.48 | 52.11 | 42.68 | 43.99 | 48.32 | 40.48 | 44.72 | 44.84 |
| | | FC-Encoder (GIN) | 43.29 | 44.69 | 40.21 | 33.79 | 54.54 | 49.56 | 52.14 | 46.97 |
| | | FC-Encoder (GCN) | 54.20 | 50.13 | 53.00 | 51.76 | 63.94 | 57.11 | 58.65 | 57.14 |
| | | Cayley-Encoder (GIN) | **55.84** | 55.97 | **60.25** | **56.65** | **66.99** | **58.63** | 56.59 | 55.34 |
| | | Cayley-Encoder (GCN) | 49.53 | **57.90** | 51.10 | 52.60 | 60.51 | 56.62 | 39.32 | 52.15 |
| Llama3 8B | Baselines | Last-Layer | 45.63 | 32.65 | 52.66 | 42.88 | 58.07 | 54.32 | 67.00 | 51.16 |
| | | Best-Layer | 56.51 | 47.21 | 60.09 | 54.62 | 66.89 | 59.04 | **72.83** | 56.96 |
| | | MLP Last-Layer | 48.91 | 34.88 | 54.80 | 44.08 | 61.19 | 52.55 | 55.35 | 46.62 |
| | | Weighted | 45.77 | 32.86 | 59.46 | 54.90 | 67.30 | 58.97 | 52.82 | 56.91 |
| | | DWAtt | 52.85 | 51.47 | 56.26 | 58.63 | 66.00 | 48.86 | 46.88 | 51.45 |
| | ILSE | Set-Encoder | 42.31 | 39.94 | 52.80 | 52.00 | 56.12 | 39.50 | 32.96 | 46.60 |
| | | FC-Encoder (GIN) | 50.87 | 41.39 | 34.04 | 37.66 | 63.27 | 53.89 | 49.62 | 51.85 |
| | | FC-Encoder (GCN) | 59.33 | 55.19 | 53.51 | 59.88 | 73.69 | 57.25 | 62.81 | 57.51 |
| | | Cayley-Encoder (GIN) | **63.05** | **65.31** | **63.53** | **70.17** | **76.96** | **63.13** | 55.32 | **60.74** |
| | | Cayley-Encoder (GCN) | 41.71 | 47.11 | 42.48 | 48.51 | 52.46 | 42.09 | 23.25 | 44.04 |

## B    ILSE SIZE VS LLM BASE MODELS

Table 3 shows the number of learned parameters learned on top of the frozen LLM for each method. For the Last-Layer and Best-Layer methods, there are no added learned parameters.

Table 3: Trainable parameters added by each method. All base model parameters remain frozen. Percentages show overhead relative to base model size.

| Method | Pythia-410M | Llama3-8B |
|---|---|---|
| *Frozen Parameters* | *410M* | *8B* |
| Weighted | 25 | 33 |
| MLP | 394K (0.10%) | 1.18M (0.015%) |
| Set-Encoder | 394K (0.10%) | 1.18M (0.015%) |
| FC-Encoder | 395K (0.10%) | 1.18M (0.015%) |
| Cayley-Encoder | 395K (0.10%) | 1.18M (0.015%) |
| DWAtt | 2.0M (0.49%) | 3.3M (0.04%) |

## C    HARDWARE DETAILS

We worked on HPC cluster with access to the following GPUs: A6000, A100, V100 and geforce rtx 3090.

## D    TRAIN TIME

We couldn't compare training time properly because our experiments ran on different GPUs, depending on the cluster availability. However we decided to add this table for rough estimates of training times among the different methods.

Table 4: Average training time for ISLE and baselines across all LLMs. Times are approximate due to GPU hardware variability (A6000, A100, V100, RTX 3090).

| Section | Method | Bank77 | Emot. | MTOP-D | MTOP-I | Poem | STS-B | Avg |
|---|---|---|---|---|---|---|---|---|
| Baselines | Weighted | 1.0m | 1.3m | 1.2m | 1.3m | 5.0s | 1.1h | 11.5m |
| | DWATT | 11.4m | 1.1h | 26.3m | 57.3m | 3.0m | 57.8m | 36.5m |
| | MLP | 11.0s | 19.1s | 19.2s | 18.3s | 1.4s | 2.8m | 39.3s |
| ILSE | Set-Encoder | 42.6s | 2.5m | 1.7m | 2.6m | 5.7s | 1.5h | 16.7m |
| | FC-Encoder (GIN) | 2.6m | 4.5m | 4.3m | 4.8m | 17.9s | 52.8m | 11.5m |
| | FC-Encoder (GCN) | 3.0m | 5.4m | 4.8m | 4.4m | 17.6s | 55.2m | 12.2m |
| | Cayley-Encoder (GIN) | 3.4m | 6.1m | 7.4m | 5.8m | 18.4s | 3.2h | 35.4m |
| | Cayley-Encoder (GCN) | 4.2m | 7.4m | 6.8m | 6.4m | 26.1s | 1.7h | 21.6m |

# E CLASSIFICATION RESULTS WITH DELTA CALCULATIONS

We include here an extended version of Table 1, which includes the delta percentages between ILSE and the baselines for each LLM. In classification all ILSE methods improve over the baselines, except for one case.

Table 5: Performance comparison of ILSE VS baselines on classification tasks with delta improvements. **Bold**: best per column, blue: 2nd best, red: 3rd best. Deltas: green = improvement, red = degradation.

| Base Model | Section | Method | Banking77 | Emotion | MTOPDomain | MTOPIntent | PoemSentiment | Avg |
|---|---|---|---|---|---|---|---|---|
| Pythia 410m | Baselines | Last Layer | 61.17 | 33.48 | 80.88 | 66.97 | 42.40 | 56.98 |
| | | Best Single Layer | 66.67 | 35.02 | 83.78 | 71.18 | 45.67 | 60.47 |
| | | MLP Last Layer | 83.84 | 33.99 | 96.97 | 83.75 | 75.00 | 74.71 |
| | | MLP Best Layer | 41.93 | 25.41 | 87.25 | 70.79 | 53.94 | 55.86 |
| | | Weighted | 58.50 | 28.26 | 79.79 | 61.68 | 42.60 | 54.16 |
| | | DWATT | 83.23 | 58.60 | 98.03 | 91.66 | 70.87 | 80.48 |
| | ILSE | Set-Encoder | 84.23 | 47.89 | 97.59 | 92.21 | 73.37 | 79.06 |
| | | FC-Encoder (GIN) | 90.10 | 73.36 | 98.68 | 94.32 | 69.90 | 85.27 |
| | | FC-Encoder (GCN) | **90.65** | **75.61** | 98.65 | **95.04** | **75.77** | 87.14 |
| | | Cayley-Encoder (GIN) | 89.12 | 73.83 | **98.77** | 94.72 | 70.87 | 85.46 |
| | | Cayley-Encoder (GCN) | 89.43 | 66.40 | 98.67 | 94.19 | 69.13 | 83.56 |
| Llama3 8B | Baselines | Last Layer | 68.25 | 34.23 | 84.42 | 73.39 | 40.96 | 60.25 |
| | | Best Single Layer | 71.93 | 38.42 | 89.01 | 78.17 | 47.02 | 64.91 |
| | | MLP Last Layer | 86.70 | 67.67 | 98.58 | 92.09 | 75.00 | 84.01 |
| | | MLP Best Layer | 49.59 | 21.17 | 47.64 | 60.04 | 35.67 | 42.82 |
| | | Weighted | 66.63 | 27.94 | 83.85 | 71.78 | 37.79 | 57.60 |
| | | DWATT | 90.04 | 66.55 | 97.97 | 92.41 | 75.00 | 84.40 |
| | ILSE | Set-Encoder | 87.62 | 71.04 | 98.77 | 95.43 | 77.02 | 85.98 |
| | | FC-Encoder (GIN) | 92.10 | 71.64 | 98.77 | 95.65 | 75.96 | 86.83 |
| | | FC-Encoder (GCN) | 92.38 | 71.03 | 98.99 | 96.19 | 77.98 | 87.31 |
| | | Cayley-Encoder (GIN) | 92.46 | 71.58 | 98.98 | 96.46 | 76.54 | 87.20 |
| | | Cayley-Encoder (GCN) | **92.85** | **73.43** | **99.03** | 95.90 | **79.04** | 88.05 |
| | Δ vs Last Layer | Set-Encoder | +21.22 | +25.61 | +15.53 | +23.64 | +33.51 | +23.90 |
| | | FC-Encoder (GIN) | +26.39 | +38.64 | +16.08 | +24.81 | +31.25 | +27.44 |
| | | FC-Encoder (GCN) | +26.81 | +39.46 | +16.17 | +25.44 | +35.19 | +28.61 |
| | | Cayley-Encoder (GIN) | +26.08 | +38.84 | +16.22 | +25.41 | +32.02 | +27.72 |
| | | Cayley-Encoder (GCN) | +26.43 | +36.06 | +16.20 | +24.87 | +32.40 | +27.19 |
| | Δ vs Best Single Layer | Set-Encoder | +16.62 | +22.74 | +11.79 | +19.15 | +28.85 | +19.83 |
| | | FC-Encoder (GIN) | +21.80 | +35.78 | +12.33 | +20.31 | +26.59 | +23.36 |
| | | FC-Encoder (GCN) | +22.21 | +36.59 | +12.42 | +20.94 | +30.53 | +24.54 |
| | | Cayley-Encoder (GIN) | +21.49 | +35.98 | +12.48 | +20.91 | +27.36 | +23.64 |
| | | Cayley-Encoder (GCN) | +21.83 | +33.19 | +12.45 | +20.37 | +27.74 | +23.12 |
| | Δ vs MLP Last Layer | Set-Encoder | +0.66 | +8.64 | +0.41 | +5.90 | +0.19 | +3.16 |
| | | FC-Encoder (GIN) | +5.83 | +21.67 | +0.95 | +7.07 | -2.07 | +6.69 |
| | | FC-Encoder (GCN) | +6.25 | +22.49 | +1.05 | +7.70 | +1.88 | +7.87 |
| | | Cayley-Encoder (GIN) | +5.52 | +21.87 | +1.10 | +7.67 | -1.30 | +6.97 |
| | | Cayley-Encoder (GCN) | +5.87 | +19.09 | +1.08 | +7.13 | -0.91 | +6.45 |
| | Δ vs DWATT | Set-Encoder | -0.71 | -3.11 | +0.18 | +1.79 | +2.26 | +0.08 |
| | | FC-Encoder (GIN) | +4.47 | +9.92 | +0.73 | +2.95 | +0.00 | +3.61 |
| | | FC-Encoder (GCN) | +4.88 | +10.74 | +0.82 | +3.58 | +3.94 | +4.79 |
| | | Cayley-Encoder (GIN) | +4.15 | +10.13 | +0.87 | +3.55 | +0.77 | +3.89 |
| | | Cayley-Encoder (GCN) | +4.50 | +7.34 | +0.85 | +3.01 | +1.15 | +3.37 |
| | Δ Cayley vs FC | Cayley-Encoder (GIN) | -0.31 | +0.20 | +0.15 | +0.60 | +0.77 | +0.28 |
| | | Cayley-Encoder (GCN) | -0.38 | -3.41 | +0.03 | -0.57 | -2.79 | -1.42 |
| | Δ Avg Cayley vs FC | Cayley - FC | -0.35 | -1.60 | +0.09 | +0.02 | -1.01 | -0.57 |

# F  STS Results with Delta Calculations

We include here an extended version of Table 2, which includes the delta percentages between ILSE and the baselines for each LLM. In STS Cayley-Encoder shows dominance among ILSE methods.

Table 6: Performance comparison of ILSE VS baselines on STS tasks with delta improvements. **Bold**: best per column, blue: 2nd best, red: 3rd best. Deltas: green = improvement, red = degradation.

| Base Model | Section | Method | STSBenchmark | STS12 | STS13 | STS14 | STS15 | STS16 | BIOSSES | SICK-R | Avg |
|---|---|---|---|---|---|---|---|---|---|---|---|
| Pythia 410m | Baselines | Last-Layer | 39.12 | 46.96 | 47.00 | 41.45 | 49.32 | 50.37 | 67.30 | 52.55 | 49.26 |
| | | Best-Layer | 53.53 | 50.62 | 59.27 | 51.61 | 65.59 | 58.02 | 74.80 | 58.26 | 58.96 |
| | | MLP Last-Layer | 25.54 | 42.14 | 36.16 | 33.28 | 37.75 | 39.84 | 59.68 | 43.23 | 39.70 |
| | | Weighted | 41.81 | 47.12 | 49.49 | 44.21 | 53.72 | 52.76 | 67.79 | 55.09 | 51.50 |
| | | DWAtt | 49.40 | 52.51 | 44.57 | 47.46 | 52.51 | 44.80 | 45.29 | 52.43 | 48.62 |
| | ILSE | Set-Encoder | 39.48 | 52.11 | 42.68 | 43.99 | 48.32 | 40.48 | 44.72 | 44.84 | 44.58 |
| | | FC-Encoder (GIN) | 43.29 | 44.69 | 40.21 | 33.79 | 54.54 | 49.56 | 52.14 | 46.97 | 45.65 |
| | | FC-Encoder (GCN) | 54.20 | 50.13 | 53.00 | 51.76 | 63.94 | 57.11 | 58.65 | 57.14 | 55.74 |
| | | Cayley-Encoder (GIN) | 55.84 | 55.97 | 60.25 | 56.65 | 66.99 | 58.63 | 56.59 | 55.34 | 58.28 |
| | | Cayley-Encoder (GCN) | 49.53 | 57.90 | 51.10 | 52.60 | 60.51 | 56.62 | 39.32 | 52.15 | 52.47 |
| Llama3 8B | Baselines | Last-Layer | 45.63 | 32.65 | 52.66 | 42.88 | 58.07 | 54.32 | 67.00 | 51.16 | 50.54 |
| | | Best-Layer | 56.51 | 47.21 | 60.09 | 54.62 | 66.89 | 59.04 | 72.83 | 56.96 | 59.27 |
| | | MLP Last-Layer | 48.91 | 34.88 | 54.80 | 44.08 | 61.19 | 52.55 | 55.35 | 46.62 | 49.80 |
| | | Weighted | 45.77 | 32.86 | 59.46 | 54.90 | 67.30 | 58.97 | 52.82 | 56.91 | 53.62 |
| | | DWAtt | 52.85 | 51.47 | 56.26 | 58.63 | 66.00 | 48.86 | 46.88 | 51.45 | 54.05 |
| | ILSE | Set-Encoder | 42.31 | 39.94 | 52.80 | 52.00 | 56.12 | 39.50 | 32.96 | 46.60 | 45.28 |
| | | FC-Encoder (GIN) | 50.87 | 41.39 | 34.04 | 37.66 | 63.27 | 53.89 | 49.62 | 51.85 | 47.82 |
| | | FC-Encoder (GCN) | 59.33 | 55.19 | 53.51 | 59.88 | 73.69 | 57.25 | 62.81 | 57.51 | 59.90 |
| | | Cayley-Encoder (GIN) | 63.05 | 65.31 | 63.53 | 70.17 | 76.96 | 63.13 | 55.32 | 60.74 | 64.77 |
| | | Cayley-Encoder (GCN) | 41.71 | 47.11 | 42.48 | 48.51 | 52.46 | 42.09 | 23.25 | 44.04 | 42.71 |
| | Δ vs Last-Layer | Set-Encoder | -1.48 | +6.22 | -2.09 | +5.82 | -1.47 | -12.36 | -28.31 | -6.14 | -4.97 |
| | | FC-Encoder (GIN) | +4.70 | +3.24 | -12.71 | -6.44 | +5.21 | -0.62 | -16.27 | -2.45 | -3.17 |
| | | FC-Encoder (GCN) | +14.39 | +12.86 | +3.42 | +13.65 | +15.12 | +4.83 | -6.42 | +5.47 | +7.92 |
| | | Cayley-Encoder (GIN) | +17.07 | +20.84 | +21.24 | +18.28 | +8.53 | -11.20 | +6.19 | +11.63 | |
| | | Cayley-Encoder (GCN) | +3.24 | +12.70 | -3.04 | +8.39 | +2.79 | -2.99 | -35.86 | -3.76 | -2.32 |
| | Δ vs Best-Layer | Set-Encoder | -14.12 | -2.89 | -11.94 | -5.13 | -14.02 | -18.54 | -34.97 | -11.89 | -14.19 |
| | | FC-Encoder (GIN) | -7.94 | -5.87 | -22.56 | -17.39 | -7.34 | -6.81 | -22.94 | -8.20 | -12.38 |
| | | FC-Encoder (GCN) | +1.74 | +3.75 | -6.43 | +2.70 | +2.58 | -1.36 | -13.08 | -0.29 | -1.30 |
| | | Cayley-Encoder (GIN) | +4.43 | +11.73 | +2.21 | +10.29 | +5.74 | +2.34 | -17.86 | +0.43 | +2.41 |
| | | Cayley-Encoder (GCN) | -9.40 | +3.59 | -12.89 | -2.56 | -9.76 | -9.18 | -42.53 | -9.52 | -11.53 |
| | Δ vs MLP Last-Layer | Set-Encoder | +3.67 | +7.52 | +2.26 | +9.31 | +2.75 | -6.21 | -18.67 | +0.80 | +0.18 |
| | | FC-Encoder (GIN) | +9.85 | +4.53 | -8.36 | -2.95 | +9.43 | +5.53 | -6.63 | +4.49 | +1.99 |
| | | FC-Encoder (GCN) | +19.54 | +14.15 | +7.77 | +17.14 | +19.34 | +10.98 | +3.22 | +12.40 | +13.07 |
| | | Cayley-Encoder (GIN) | +22.22 | +22.13 | +16.41 | +24.73 | +22.50 | +14.68 | -1.56 | +13.12 | +16.78 |
| | | Cayley-Encoder (GCN) | +8.39 | +14.00 | +1.31 | +11.88 | +7.01 | +3.16 | -26.22 | +3.17 | +2.84 |
| | Δ vs DWAtt | Set-Encoder | -10.23 | -5.97 | -2.67 | -5.05 | -7.03 | -6.84 | -7.24 | -6.22 | -6.41 |
| | | FC-Encoder (GIN) | -4.05 | -8.95 | -13.29 | -17.32 | -0.36 | +4.89 | -2.53 | -4.60 | |
| | | FC-Encoder (GCN) | +5.64 | +0.67 | +2.84 | +2.77 | +9.56 | +10.35 | +14.65 | +5.39 | +6.48 |
| | | Cayley-Encoder (GIN) | +8.32 | +8.65 | +11.48 | +10.37 | +12.72 | +14.05 | +9.87 | +6.10 | +10.19 |
| | | Cayley-Encoder (GCN) | -5.51 | +0.51 | -3.62 | -2.49 | -2.77 | +2.52 | -14.79 | -3.84 | -3.75 |
| | Δ Cayley vs FC | Cayley-Encoder (GIN) | +12.37 | +17.60 | +24.77 | +27.68 | +13.07 | +9.16 | +5.07 | +8.63 | +14.79 |
| | | Cayley-Encoder (GCN) | -11.15 | -0.16 | -6.46 | -5.26 | -12.33 | -7.82 | -29.44 | -9.23 | -10.23 |

