# OpenReview forum: "Improving LLM Predictions via Inter-Layer Structural Encoders"
_ICLR.cc/2026/Workshop/GRaM — ICLR 2026 Workshop GRaM Poster_

### Official Review · Reviewer_Uv61 · 2026-02-15
**Structured iInter-Layer aggregation via Cayley Graphs for frozen LLM representations**

**Rating:** 8
**Confidence:** 4

**Review:**

This paper proposes Inter-Layer Structural Encoders (ILSE), a framework for aggregating representations from all layers of a frozen LLM. The main contribution is the Cayley-Encoder, which maps layer representations to nodes in a 4-regular Cayley graph to enable structured message passing. Experiments across 13 MTEB classification and STS tasks and three LLMs (410M–8B) show consistent improvements over last-layer, best-layer, weighted averaging, MLP, and depth-wise attention baselines, including strong few-shot performance.
Strengths:
- Consistent empirical gains across models and tasks, particularly in classification and few-shot settings.
- Well-motivated geometric approach, leveraging expander properties of Cayley graphs to improve inter-layer communication.
- Parameter-efficient: adds only ~0.015–0.1% overhead relative to base models.
- Clean experimental setup with frozen backbones and multiple baselines.

Weaknesses:
- Novelty lies mainly in the choice of graph topology; conceptually, it remains a GNN-based aggregator over frozen embeddings.
- Limited ablation on graph structure (e.g., comparison to other expander or random regular graphs).
- Some instability in STS results across variants (e.g., GIN vs GCN), with limited analysis of variance.
- Lacks interpretability analysis showing how different layers contribute after aggregation.

**Pmlr Suitability:**

Yes

---

### Official Review · Reviewer_eV7U · 2026-02-23
**Review for "Improving LLM predictions via inter-layer structural encoders"**

**Rating:** 6
**Confidence:** 3

**Review:**

This paper modifies a transformer architecture by allowing the output layer to receive activations from all intermediate layers principally via a GNN architecture, a method the authors call Inter-Layer Structural Encoders (ILSE). They benchmark several variants, based on Cayley graps and complete graphs, and investigating both GCN and GIN aggregation functions.

As far as I am aware using GNN message passing methods to aggregate activations for the output layer is novel. The results given in the experimental section are strong, and the paper is well written, although some details are omitted that I would have like to have seen. Overall I think the paper could be accepted to the workshop and would be a nice addition to the programme. I think the theme of the paper (using geometrical methods to improve transformer performance) is suitably relevant to the GRaM workshop topic.

Here are a number of further comments.

1. There is no information that I can see about the number of runs, or the standard deviation, either in the main body or the appendix. This is a significant weakness.

2. Message passing - what is the structure of this? E.g. how many rounds of message passing are performed? If it's 1 round vs several that fundamentally changes the interpretation. Omission of this information is also a significant weakness.

3. The bottleneck problem for graph message passing algorithms is mentioned several times, but I don't see that as very relevant or well-explored. Both the Cayley and FC graph architectures studied will not suffer from bottlenecks. The number of layers is low (~20) meaning that bottleneck are unlikely to be a problem anyway. Whatever phenomena is happening here, I don't think it is to do with bottlenecks. So, why exactly is this method so effective? It is disappointing not to even see speculation about this.

4. On STS tasks, the authors find that the Cayley method outperforms the FC method. I think this is quite surprising. There aren't very many layers (~20). We don't have any standard deviations so we can't judge if this difference is significant. Some sort of ablation/investigation would be very welcome.

5. It is good to see the parameter count values clearly set out in the appendix, which show that this method only adds ~0.1% to the existing network parameter count.

6. Why SL(2,Z)?

7. Set-Encoder - it seems something of a distraction to include this architecture, as it is quite different to the others, and underperforms compared to them.

8. Table 5 - I think these deltas are the average across each LLM, this wasn't super clear and you could explain this in the table heading.

**Pmlr Suitability:**

NA

---

### Official Review · Reviewer_xzeT · 2026-02-24
**Review for "Improving LLM Predictions via Inter-Layer Structural Encoders"**

**Rating:** 6
**Confidence:** 3

**Review:**

***Summary***

This paper argues that using only a frozen LLM’s final-layer embedding is often not the best choice for downstream tasks. The authors propose Inter-Layer Structural Encoders (ILSE): take the embedding from every layer, treat layers as nodes and learn an aggregation function to produce a single representation. Three variants are explored: Set-Encoder, FC-Encoder, and the proposed Cayley-Encoder. Across the classification results and the STS transfer setup, ILSE generally improves over common baselines (Last-Layer, Best-Layer, weighted aggregation, etc) while keeping the added trainable component relatively small.

I think this paper is a good fit because it introduces an  geometric(graph-structured) layer-aggregation method (via a Cayley graph) to improve representations from models, which aligns with the workshop’s focus on geometry-informed methods at scale.

***Strengths***

* The gains over baselines look significant.
* Few-shot results look strong, good when data is limited.
* The base LLM stays frozen, and the extra encoder is lightweight in trainable parameters

***Discussions***
1. In a few tasks, MLP best-layer is worse than Best-Layer/Last-Layer, which is unexpected to me. Could the authors clarify how Best-Layer is selected (eg. which split, per task/model, fixed across runs), and whether the same selection/tuning procedure is used for the MLP baseline?
2. The “over-squashing and bottleneck-free” story for Cayley is interesting, but it is mostly supported by end-task results. Any simple control (eg, a degree-matched graph) to explore whether Cayley itself is the reason?
3. Table 1 suggests FC-Encoder can beat Cayley in some settings Any intuition on when dense layer-to-layer mixing is better?
4. What do virtual nodes do in practice?
5. Since layers are randomly mapped to Cayley nodes, how sensitive are results to that randomness?
6. For STS, which correlation metric is reported?

**Pmlr Suitability:**

NA

---

### Meta-Review · Area_Chair_cGkQ · 2026-02-25

**Decision:**

Accept

**Metareview:**

This tiny paper presents a novel, well-motivated idea for aggregating information across layers. The reviewers found the results correct, and the experiments strong. The topic is also relevant for GRaM. Thus, I recommend for acceptance.

**Relevance To Proceedings:**

Tiny paper — does not apply

**Relevance To Workshop:**

Yes — suitable for GRaM

---

### Decision · Program_Chairs · 2026-03-02

Accept (Poster)